# CondiQuant: Condition Number Based Low-Bit Quantization for Image Super-Resolution

## Abstract

Low-bit model quantization for image super-resolution (SR) is a longstanding task that is renowned for its surprising compression and acceleration ability. However, accuracy degradation is inevitable when compressing the full-precision (FP) model to ultra-low bit widths ($2 \sim 4$ bits). Experimentally, we observe that the degradation of quantization is mainly attributed to the quantization of activation instead of model weights. In numerical analysis, the condition number of weights could measure how much the output value can change for a small change in the input argument, inherently reflecting the quantization error. Therefore, we propose **CondiQuant**, a **condi**tion number-based low-bit post-training **quant**ization for image super-resolution. Specifically, we formulate the quantization error as the condition number of weight metrics. By decoupling the representation ability and the quantization sensitivity, we design an efficient proximal gradient descent algorithm to iteratively minimize the condition number and maintain the output. With comprehensive experiments, we demonstrate that CondiQuant outperforms existing state-of-the-art post-training quantization methods in accuracy without computation overhead and gains the theoretically optimal compression ratio in model parameters. Our code will be released soon.

## 1 Introduction

Image super-resolution (SR) aims to restore the high-resolution (HR) images from the low-resolution counterparts. It is a foundational computer vision task in low-level vision and image processing, widely studied in medical imaging (Greenspan, 2008; Isaac & Kulkarni, 2015; Huang et al., 2017), surveillance (Zhang et al., 2010; Rasti et al., 2016), remote sensing (Bandara & Patel, 2022), and mobile phone photography (Wu et al., 2024). Nonetheless, the existing edge devices' notorious limited computation and memory ability hinder real-world deployment. Therefore, it is increasingly urgent to develop model compression and acceleration techniques for SR models to reduce the redundancy in both model parameters and inference computation.

Model quantization (Choukroun et al., 2019; Ding et al., 2022; Hubara et al., 2021; Li et al., 2021) is a powerful compression technique that compresses

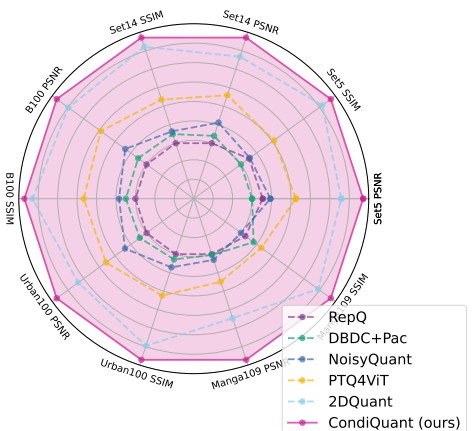

Figure 1: Comparison with SOTA PTQ methods on five benchmarks. Our CondiQuant gains consistently better performance.

the model from full-precision to low-bit representations. With quantization, the time-consuming floating-point operations are converted into efficient integer ones, making it an ideal candidate for model compression in resource-constrained edge devices. However, the conversion inevitably leads to severe performance degradation, especially when compressing to ultra-low bit width ($2 \sim 4$ bits). The situation is more severe in vision transformers (ViTs) due to the deterioration of self-attention.

The two branches of quantization, aka quantization-aware training (QAT) and post-training quantization (PTQ), deal with the degradation in different ways. QAT, a firm adherent of backpropagation,

concurrently optimizes the model weights and quantizers' parameters via straight-through estimator (STE) (Courbariaux et al., 2016). Though inspiring results are obtained, the ineluctable demand for large datasets and long-time training discourages the deployment. Whereas PTQ is much more efficient. Most PTQ methods utilize efficient algorithms to calculate the quantizers' parameters in a small calibration set. Given the complexity of both branches, PTQ is usually inferior to QAT and calls for advanced solutions. To improve the inferior performance of PTQ, we observe two key experimental phenomena. The first observation is about the attribution of degradation.

> ***Observation 1:*** *The degradation of quantization is mainly attributed to **activations**.*

Therefore, we leverage condition number, a metric in numerical analysis, to reflect the changes caused by the quantization of activations. In detail, condition number could measure how much the output value can change given a disturbance in the input value, *i.e.,* the quantization error of activation. Considering a small disturbance, the smaller the condition number is, the smaller the output varies. Thereafter, one way to reduce the quantization error is by minimizing the condition number of the weight matrix. Inspired by MagR (Zhang et al., 2024), we verify that their observation in LLMs also holds in SR models, which is

> ***Observation 2:*** *The feature matrix $X$ across all linear layers is approximately **rank deficient**.*

In a linear layer, the output $Y$ can be expressed as matrix multiplication $Y = XW$, where $W$ is the weight matrix and $X$ is the feature matrix. When $X$ is rank-deficient, there are infinite solutions for $W$ that satisfy the equation. This desirable property allows a shift in the weight matrix to quantization friendliness while maintaining the output.

Based on the previous two observations, we focus on the impact of quantization of activation, establish its relationship with condition number, formulate the optimization problem, and derive proximal gradient descent to solve it efficiently. With the proposed CondiQuant, the restoration accuracy is improved evidently as shown in Fig. 1.

To sum up, our main contributions are fourfold:

1. We design CondiQuant, a novel PTQ method for image SR based on condition number. CondiQuant gains theoretically optimal compression and speedup ratios with the currently minimum performance degradation.

2. We propose that the model's sensitivity to quantization is related to the condition number of weights in the linear layers and could be optimized efficiently.

3. We design an efficient algorithm based on proximal gradient descent to reduce the condition number of the weights while approximately maintaining the output.

4. Comprehensive comparison experiments are conducted to show the SOTA performance on efficiency and effectiveness of our proposed CondiQuant. Besides, extensive ablation studies are conducted to prove the robustness and efficacy of our proposed CondiQuant.

## 2 RELATED WORK

**Image Super-Resolution.** The trailblazing research on image super-resolution with deep neural network is SRCNN (Dong et al., 2016), which utilizes convolution layers to replace conventional methods in SR. Since then, dazzling network architectures (Chen et al., 2022; 2023; Lim et al., 2017; Zhang et al., 2018a) are designed to achieve more stirring image restoration accuracy. RDN (Zhang et al., 2018b) designs dense skip connection and global residual to utilize abundant local and global features. With the advance of vision transformer, SwinIR (Liang et al., 2021) uses window-attention to fully make use of global information. As reconstruction accuracy increases, researchers focus more on model parameters and operations. Generally, the ViT-based SR networks are much smaller and faster than pure CNNs. However, even the advanced SR networks like SwinIR, current SR models are still too large to be deployed on resource-constrained edge devices. Therefore, research of quantization on ViT-based networks is in urgent need to make SR possible on mobile devices.

**Post-Training Quantization.** PTQ is famous for its fast speed and low cost during quantization. Recently, excellent PTQ methods have been proposed to advance the performance and efficiency of

PTQ. As a PTQ method specifically for ViT, PTQ4ViT (Yuan et al., 2022) proposed twin uniform quantization to reduce quantization error on the output of softmax and GELU. RepQ (Li et al., 2023) decouples the quantization and inference processes and ensures both accurate quantization and efficient inference. NoisyQuant (Liu et al., 2023) surprisingly finds that adding a fixed Uniform noisy bias to the values being quantized can significantly reduce the quantization error. However, most of the above PTQ methods are only for Transformer blocks and the generalization ability on SR tasks is relatively low. Whereas, 2DQuant (Liu et al., 2024) searches the clipping bounds in different ways on different distributions and is currently the best PTQ method on the SR task. However, most of the above methods only concentrate on the quantization process and ignore adjustments in the weights.

**Condition Number.** The concept of the condition number originated and developed in numerical analysis (Turing, 1948; von Neumann & Goldstine, 1947). It measures how much the output value of the system or function can change for a small change in the input. A matrix with a high condition number is said to be ill-conditioned and leads to huge output changes given a small disturbance in input. It also plays an important role in machine learning. Freund proposed optimization algorithms generally converge faster when the condition number is low (Freund et al., 2018). In causal inference, it is used to evaluate the numerical stability of causal effect estimation (Gordon et al., 2021). Additionally, the condition number is leveraged to enhance the numerical stability of regression models by fine-tuning hidden layer parameters (Xiao et al., 2018). In this work, we formulate the optimization of quantization error via condition number and propose CondiQuant to solve it efficiently.

## 3 METHOD

### 3.1 PRELIMINARIES

Given an SR network, we use upper and lower clipping bounds to quantize weights and activations (Liu et al., 2024). Usually, we leverage the fake quantization to simulate the quantization process with no simulation error. The fake-quantize process can be written as:

$$x_c = \text{Clip}(x, l, u), x_r = \text{R}(s(x_c - l)), x_q = sx_r + l, \tag{1}$$

where $x$ denotes weight $w$ or activation $a$ being quantized and $s = (2^N - 1)/(u - l)$ is the quantization ratio. $l$ and $u$ denote the lower bound and upper bound for clipping, respectively. $\text{Clip}(x, l, u) = \min(\max(x, l), u)$ constrains the input to be between $l$ and $u$, and $\text{R}(\cdot)$ rounds the input to nearest integer. With the above, the continuous values are dispersed into discrete candidates.

### 3.2 ANALYSIS

To analyze the loss attribution in quantization, we rewrite the linear layer with quantization as follows:

$$\hat{Y} := \hat{X}\hat{W} = (X + \delta X)(W + \delta W) = XW + X\delta W + \delta X W + \delta X \delta W,$$

where $\hat{X}$ denotes the quantized value and $\delta X$ denotes the quantization error of $X$. The second-order term $\delta X \delta W$ can be ignored due to its tiny impact on performance. Thus Eq. 2 can be rewritten as:

$$\hat{Y} \approx XW + X\delta W + \delta X W = Y + X\delta W + \delta X W. \tag{2}$$

Shown in Tab. 1, we observe that the influence of $X\delta W$ and $\delta X \delta W$ is smaller than $\delta X W$.

This result can be explained by two reasons. First, the activation varies with different inputs while the weight is fixed. Therefore, the average value of the absolute value of the $\delta X$ elements is much greater than that of $\delta W$. Second, the $W$ is sensitive to minor changes, *i.e.,* $\delta X$. Hence, we only concentrate on $\delta X W$ and ignore $X\delta W$ in the following discussion. So we can further approximate the loss:

|  | $Y$ | $Y + X\delta W$ | $Y + \delta X W$ | $Y + \delta X \delta W$ |
|---|---|---|---|---|
| PSNR | 32.2543 | 32.0117 | 31.3304 | 32.2519 |
| SSIM | 0.9293 | 0.9270 | 0.9221 | 0.9293 |

Table 1: Attribution of quantization loss. $Y$ denotes the FP model and the rest denotes performing quantization on $W$, $X$, and both.

$$\hat{Y} \approx XW + \delta X W, \quad \delta Y \approx \delta X W. \tag{3}$$

3 indicates that the degradation of model performance can be divided into two components: (1) the magnitude of activation quantization error and the sensitivity of the weight to $\delta$. To minimize the approximate loss, a naive solution is to reduce the Frobenius norm of $\delta X$ *i.e.,* $||\delta X||_F$. However, it is impractical as $\delta X$ is input-related, and rounding-off errors are impossible to reduce, especially with low bits. Therefore, we focus on the model's sensitivity to quantization, related to condition number.

### 3.3 CONDIQUANT

We begin with the observation of rank deficiency of the feature matrix. Then we derive the optimization objective and illustrate the methodology to arrive at the solution.

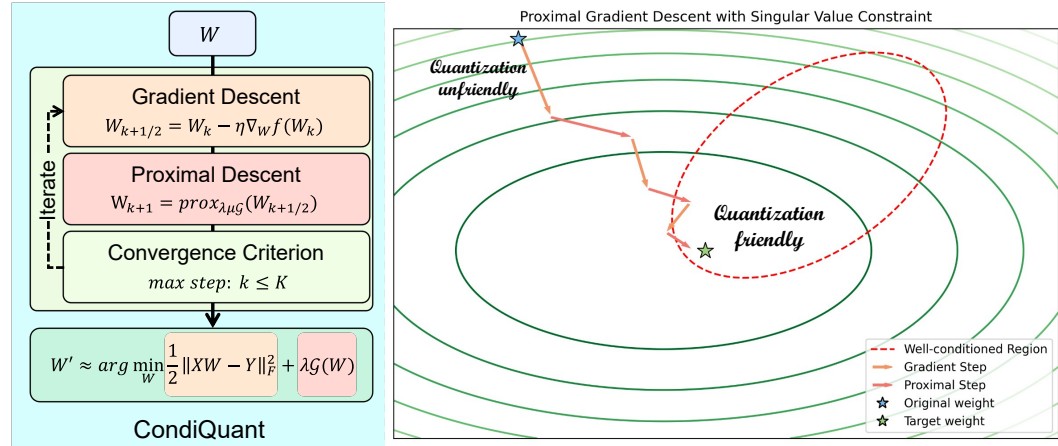

Figure 3: Overview of our proposed CondiQuant, which employs iterative optimization to minimize the condition number of the weight matrix while maintaining the output. The gradient descent step updates the weight matrix with gradients to ensure the output is close to the original. The proximal descent step minimizes the condition number with the proximal operator. The above two steps are conducted iteratively before reaching the convergence criterion, *i.e.*, the max iteration step. We illustrate the effect of both steps. With CondiQuant, the weight matrix is converted into a quantization-friendly and well-conditioned one, while the output is close to the original.

**Rank Deficiency.** We visualize the rank ratio in Eq. 2. Most of the layers are rank-deficient, and only two layers are full rank. The mean value of the rank ratio is 0.626. In the linear layer, the process is matrix multiplication, *i.e.*, $Y = XW$. If $W$ is rank deficient, there exist infinite $W'$ that satisfies $XW = XW'$. Hence, this result serves as a theoretical foundation to select $W'$ to minimize $\delta XW$.

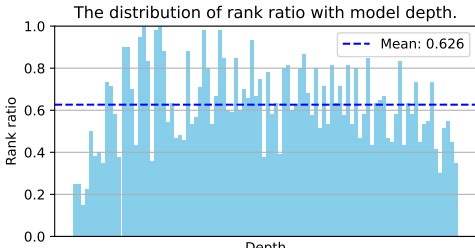

Figure 2: Distribution of activation ranks along the ($\times 2$) model depth. Most activations are severely rank-deficient.

**Formulation.** In this section, we formulate the relationship between model sensitivity and condition number. We begin with $\|\delta Y\|_2$ and $\|Y\|_2$:

$$\|\delta Y\|_2 = \|\delta XW\|_2 \leq \|\delta X\|_2 \|W\|_2, \quad \|Y\|_2 = \|XW\|_2 \geq \|X\|_2 \, \sigma_{min}(W), \quad (4)$$

where $\|\cdot\|_2$ is the bi-norm of the matrix, and $\sigma_{min}(\cdot)$ denotes the minimum singular value. With Eq. 4, we can establish the relationship between quantization loss and condition number:

$$\frac{\|\delta Y\|_2}{\|Y\|_2} \leq \frac{\|\delta X\|_2 \|W\|_2}{\|X\|_2 \, \sigma_{min}(W)} = \kappa(W) \frac{\|\delta X\|_2}{\|X\|_2}, \quad (5)$$

where $\kappa(W)$ denotes the condition number of $W$. This formula shows that under the same rounding-off error, the impact of quantization is greater with larger $\kappa(W)$. Detailed derivation can be found in the supplementary materials. Therefore, we minimize the condition number of $W$ and maintain the output. The objective can be written as:

$$\min_{W} \kappa(W) = \frac{\sigma_{\max}(W)}{\sigma_{\min}(W)}, \text{s.t.} \|XW - X\hat{W}\|_F \leq \epsilon, \quad (6)$$

where $\epsilon$ is a small positive number that limits the magnitude of the loss. However, it is a non-convex and non-smooth optimization when minimizing the condition number directly. To address this, we construct a proxy objective to optimize and propose the proximal gradient method to solve it.

**Proxy objective function.** To minimize the condition number, our strategy is to compact the distribution of all singular values as densely as possible. Specifically, we design a regularization term to minimize the deviation of singular values:

$$\min_{W} \frac{1}{2} \|XW - Y\|_F^2 + \lambda \cdot \mathcal{G}(W), \quad (7)$$

where $\mathcal{G}(\cdot)$ is a regularization term used to compact the singular value distribution and $\lambda$ is the regularization strength parameter. There are several forms of $\mathcal{G}(W)$ to compact the singular values' distribution. Considering efficiency and effect, we choose the following form:

$$\mathcal{G}(W) := \sum_{i=1}^{r} (\sigma_i(W) - t)^2, \tag{8}$$

where $r = rank(W)$ represents the number of singular values and $t$ is a value between $\sigma_{min}(W)$ and $\sigma_{max}(W)$. The median or the mean of the singular values are possible candidates for $t$ and we choose the mean value form in CondiQuant. With this form, we could provide a closed solution to minimize $\mathcal{G}(W)$ in the following section.

**Proximal Operator for $\mathcal{G}(W)$.** We utilize the proximal operator to minimize the $\mathcal{G}(W)$ part. We define the proximal operator of function $\mathcal{G}(W)$ as follows:

$$\text{prox}_{\lambda\mu\mathcal{G}}(W) := \arg\min_{Z} \left\{ \frac{1}{2}\|Z - W\|_F^2 + \lambda\mu\mathcal{G}(Z) \right\}, \tag{9}$$

where $\mu$ is a hyper-parameter to balance the optimization. The target of Eq. 9 is to minimize $\mathcal{G}(Z)$ and keep $Z$ close to $W$. To solve this problem, we perform singular value decomposition (SVD) on $W$ to extract the singular values:

$$W = U\Sigma_W V^T, \tag{10}$$

where $U$ and $V$ are orthogonal matrices and $\Sigma_W = \text{diag}(\sigma_i(V))$ is the diagonal matrix of singular values. The problem could be reduced to optimize each singular value $\sigma_i(W)$ independently and can be written as:

$$\sigma_i^*(W) := \arg\min_{\sigma_i^*} \left\{ \frac{1}{2}(\sigma_i^* - \sigma_i(W))^2 + \lambda\mu(\sigma_i^* - t)^2 \right\}. \tag{11}$$

We notice that Eq. 11 is a quadratic minimization problem, and a closed solution could be derived. Specifically, after taking the derivative and setting it to zero, we have:

$$\sigma_i^*(W) = \frac{\sigma_i(W) + 2\lambda\mu t}{1 + 2\lambda\mu}. \tag{12}$$

Using the updated singular values $\sigma_i^*(W)$, we reconstruct the matrix with lower condition number:

$$W^* = U\Sigma_{W^*} V^T, \quad \Sigma_{W^*} = \text{diag}(\sigma_i^*(W)). \tag{13}$$

To conclude, with the proximal operator, we could update $W$ to $W^*$ to minimize $\mathcal{G}(W)$. Hereby, the condition number of $W$ is reduced and the output is kept still.

**Proximal Gradient Descent.** As discussed previously, the optimization problem contains both smooth and non-smooth components. More specifically, $\frac{1}{2}\|XW - Y\|_F^2$ is a convex and differentiable function, while $\mathcal{G}(W)$ is a convex but non-differentiable function. With the proximal operator, we design the proximal gradient descent method to solve two components iteratively:

**Step1. Gradient Descent.** Update the smooth part $\frac{1}{2}\|XW - Y\|_F^2$ with the gradient descent:

$$W_{k+\frac{1}{2}} = W_k - \eta \nabla_W \frac{1}{2}\|XW - Y\|^2 \Big|_{W=W_k}, \tag{14}$$

where $\eta > 0$ is the step size, and $\nabla_W$ is the gradient of $\frac{1}{2}\|XW - Y\|_F^2$ at $W_k$. With this step, we ensure the output is as consistent as possible with the original value.

**Step2. Proximal Step.** Apply the proximal operator to the non-smooth part $\mathcal{G}(W)$ with Eq. 10, 12 and 13 to minimize the condition number:

$$W_k = \text{prox}_{\lambda\mu\mathcal{G}}(W_{k+\frac{1}{2}}) = U\Sigma_{W^*} V^T.$$

**Step3. Repeat.** Iterate the above two steps until reaching the convergence criterion, *i.e.*, $k \geq K$. $k$ is the iteration step while $K$ is the total step of iteration. We select a desired $K$ such that the difference between consecutive iterates is sufficiently small while the iteration is not too long.

We illustrate our proposed CondiQuant in Fig. 3. With the proposed CondiQuant, we convert the weight metric from the quantization-unfriendly position into a friendly position. Besides, both gradient descent and proximal descent steps are calculation efficient and the cost is inexpensive. Moreover, we introduce no additional module. Therefore, no computation and storage overhead is caused during the inference stage and we obtain the theoretic optimal compression and speedup ratio in post-training quantization. After the conversion, we leverage the previous work's scheme (Liu et al., 2024) to perform quantization and calibration with modification, and the details are described in the supplementary materials.

| $\eta$ | $10^{-1}$ | $10^{-2}$ | $10^{-3}$ | $10^{-4}$ |
|---|---|---|---|---|
| PSNR↑ | 37.13 | 37.15 | 37.15 | 37.08 |
| SSIM↑ | 0.9568 | 0.9568 | 0.9567 | 0.9564 |

(a) Ablation study on learning rate $\eta$.

| $t$ | Mean value | Median value | $(\sigma_{max} + \sigma_{min})/2$ |
|---|---|---|---|
| PSNR↑ | 37.15 | 37.12 | 37.11 |
| SSIM↑ | 0.9567 | 0.9566 | 0.9567 |

(b) Ablation study on target value $t$.

| $\lambda$ | 0.001 | 0.002 | 0.003 | 0.004 | 0.005 | 0.010 |
|---|---|---|---|---|---|---|
| PSNR↑ | 37.13 | 37.12 | 37.15 | 37.13 | 37.10 | 37.08 |
| SSIM↑ | 0.9567 | 0.9566 | 0.9567 | 0.9567 | 0.9566 | 0.9565 |

(c) Ablation study on Regularization coefficient $\lambda$.

| | 2DQuant | w/o ProxGD | w/ ProxGD |
|---|---|---|---|
| PSNR↑ | 36.00 | 37.08 | 37.15 |
| SSIM↑ | 0.9497 | 0.9557 | 0.9567 |

(d) Ablation study on ProxGD.

Table 2: Ablation studies of CondiQuant on Set5 ($\times 2$, 2 bits). The selection of hyperparameters and the effect of CondiQuant are evaluated. The results demonstrate that CondiQuant is robust and improves performance.

## 4 EXPERIMENTS

To exhibit the outstanding performance of our proposed CondiQuant, comparisons with SOTA methods are also provided in both quantitative and qualitative forms. Extensive ablation studies are conducted to show the robustness and effectiveness of our elaborate designs.

### 4.1 EXPERIMENTAL SETTINGS

**Data and Evaluation.** We employ DF2K (Timofte et al., 2017; Lim et al., 2017) as the calibration set, which combines DIV2K (Timofte et al., 2017) and Flickr2K (Lim et al., 2017). During calibration, we only use low-resolution ones and the FP model. Thereafter, CondiQuant is tested on five commonly used benchmarks in the SR field, including Set5 (Bevilacqua et al., 2012), Set14 (Zeyde et al., 2010), B100 (Martin et al., 2001), Urban100 (Huang et al., 2015), and Manga109 (Matsui et al., 2017). The evaluation metrics include PSNR and SSIM (Wang et al., 2004), which are calculated on the Y channel of the YCbCr space.

**Implementation Details.** The backbone of CondiQuant is SwinIR-light (Liang et al., 2021), a small and efficient ViT for image restoration. The scale factors include $\times 2$, $\times 3$, and $\times 4$, and bit-width includes 2, 3, and 4 bits. During condition number optimization, the calibration set size is 100 images, randomly selected from DF2K and cropped to $3 \times 64 \times 64$. We set step size $\eta = 10^{-2}$, regularization coefficient $\lambda = 0.003$, $\mu = 1$, and max iteration $K = 50$. The target value $t$ is the mean value of singular values. Our code is written with PyTorch (Paszke et al., 2019) and runs on an NVIDIA A800-80GB GPU.

### 4.2 ABLATION STUDY

In this section, we conduct four ablation studies on Set5 ($\times 2$) with 2 bits to evaluate our design.

**Learning Rate $\eta$.** In CondiQuant, $\eta$ decides the step size of the gradient descent step. We evaluate $\eta$ from $10^{-4}$ to $10^{-1}$. As shown in Tab. 2a, the impact of $\eta$ is minor in the evaluation range. We attribute this minor impact to the rank deficiency across the network and the design of proximal descent. Rank deficiency allows multiple candidates for $W$ while the SVD reconstruction also guarantees the output remains the same. Therefore, the difference between $W_{k+\frac{1}{2}}$ and $W_k$ is small enough. However, the gradient step is not unnecessary as it ensures the output won't change much.

**Selection of $t$.** As discussed in Sec.3.3, there are several ways to select the target value $t$ in $\mathcal{G}(W)$, including the mean value and median value of all singular values and $(\sigma_{min} + \sigma_{max})/2$. We try these variants and the result is shown in Tab. 2b. The stable performance with different variants indicates that, as long as the singular values are compacted, the performance could therefore be improved. So we choose the mean value of the singular values to assign $t$.

**Regularization Coefficient $\lambda$.** As shown in Tab. 2c, we test different $\lambda$ from 0.001 to 0.01. The optimal results could be obtained when $\lambda$ is set to 0.003, while others provide well-enough results.

**CondiQuant.** As shown in Tab. 2d, the existence of CondiQuant is impactful. 2DQuant provides relatively inferior results, while our modification improves the performance significantly. Besides, CondiQuant could ease the quantization difficulty and further improve the performance. Moreover, CondiQuant takes less than 19.0 seconds to perform and the computation overhead is minor. Hence, CondiQuant could serve as an efficient pre-processing technique before model quantization.

Table 3: Quantitative comparison with SOTA methods. The first and second highest methods are marked with **red** and **blue** respectively. Our proposed CondiQuant remarkably outperforms **all** other methods in **all** settings on **all** benchmarks.

| Method | Bit | Set5 (×2) | | Set14 (×2) | | B100 (×2) | | Urban100 (×2) | | Manga109 (×2) | |
|---|---|---|---|---|---|---|---|---|---|---|---|
| | | PSNR↑ | SSIM↑ | PSNR↑ | SSIM↑ | PSNR↑ | SSIM↑ | PSNR↑ | SSIM↑ | PSNR↑ | SSIM↑ |
| SwinIR-light (Liang et al., 2021) | 32 | 38.15 | 0.9611 | 33.86 | 0.9206 | 32.31 | 0.9012 | 32.76 | 0.9340 | 39.11 | 0.9781 |
| Bicubic | 32 | 32.25 | 0.9118 | 29.25 | 0.8406 | 28.68 | 0.8104 | 25.96 | 0.8088 | 29.17 | 0.9128 |
| DBDC+Pac (Tu et al., 2023) | 4 | 37.18 | 0.9550 | 32.86 | 0.9106 | 31.56 | 0.8908 | 30.66 | 0.9110 | 36.76 | 0.9692 |
| PTQ4ViT (Yuan et al., 2022) | 4 | 37.43 | 0.9571 | 33.19 | 0.9139 | 31.84 | 0.8950 | 31.54 | 0.9212 | 37.59 | 0.9735 |
| RepQ (Li et al., 2023) | 4 | 37.89 | 0.9600 | 33.47 | 0.9174 | 32.08 | 0.8975 | 31.98 | 0.9269 | 38.37 | 0.9763 |
| NoisyQuant (Liu et al., 2023) | 4 | 37.50 | 0.9570 | 33.06 | 0.9101 | 31.73 | 0.8936 | 31.31 | 0.9181 | 37.47 | 0.9723 |
| 2DQuant (Liu et al., 2024) | 4 | 37.87 | 0.9594 | 33.41 | 0.9161 | 32.02 | 0.8971 | 31.84 | 0.9251 | 38.31 | 0.9761 |
| CondiQuant (ours) | 4 | 38.03 | 0.9605 | 33.50 | 0.9180 | 32.16 | 0.8993 | 32.03 | 0.9282 | 38.57 | 0.9769 |
| DBDC+Pac (Tu et al., 2023) | 3 | 35.07 | 0.9350 | 31.52 | 0.8873 | 30.47 | 0.8665 | 28.44 | 0.8709 | 34.01 | 0.9487 |
| PTQ4ViT (Yuan et al., 2022) | 3 | 36.49 | 0.9510 | 32.49 | 0.9045 | 31.27 | 0.8854 | 30.16 | 0.9027 | 36.41 | 0.9673 |
| RepQ (Li et al., 2023) | 3 | 35.06 | 0.9325 | 31.29 | 0.8719 | 30.04 | 0.8512 | 29.17 | 0.8726 | 34.89 | 0.9518 |
| NoisyQuant (Liu et al., 2023) | 3 | 35.32 | 0.9334 | 31.88 | 0.8911 | 30.73 | 0.8710 | 29.28 | 0.8835 | 35.30 | 0.9537 |
| 2DQuant (Liu et al., 2024) | 3 | 37.32 | 0.9567 | 32.85 | 0.9106 | 31.60 | 0.8911 | 30.45 | 0.9086 | 37.24 | 0.9722 |
| CondiQuant (ours) | 3 | 37.77 | 0.9594 | 33.21 | 0.9151 | 31.94 | 0.8966 | 31.18 | 0.9197 | 38.01 | 0.9755 |
| DBDC+Pac (Tu et al., 2023) | 2 | 34.55 | 0.9386 | 31.12 | 0.8912 | 30.27 | 0.8706 | 27.63 | 0.8649 | 32.15 | 0.9467 |
| PTQ4ViT (Yuan et al., 2022) | 2 | 33.25 | 0.8923 | 30.22 | 0.8402 | 29.21 | 0.8066 | 27.31 | 0.8111 | 32.75 | 0.9093 |
| RepQ (Li et al., 2023) | 2 | 31.65 | 0.8327 | 29.19 | 0.7789 | 28.27 | 0.7414 | 26.56 | 0.7455 | 30.46 | 0.8268 |
| NoisyQuant (Liu et al., 2023) | 2 | 30.13 | 0.7620 | 28.80 | 0.7556 | 28.26 | 0.7421 | 26.68 | 0.7627 | 30.40 | 0.8204 |
| 2DQuant (Liu et al., 2024) | 2 | 36.00 | 0.9497 | 31.98 | 0.9012 | 30.91 | 0.8810 | 28.62 | 0.8819 | 34.40 | 0.9602 |
| CondiQuant (ours) | 2 | 37.15 | 0.9567 | 32.74 | 0.9103 | 31.55 | 0.8912 | 29.96 | 0.9047 | 36.63 | 0.9713 |

| Method | Bit | Set5 (×3) | | Set14 (×3) | | B100 (×3) | | Urban100 (×3) | | Manga109 (×3) | |
|---|---|---|---|---|---|---|---|---|---|---|---|
| | | PSNR↑ | SSIM↑ | PSNR↑ | SSIM↑ | PSNR↑ | SSIM↑ | PSNR↑ | SSIM↑ | PSNR↑ | SSIM↑ |
| SwinIR-light (Liang et al., 2021) | 32 | 34.63 | 0.9290 | 30.54 | 0.8464 | 29.20 | 0.8082 | 28.66 | 0.8624 | 33.99 | 0.9478 |
| Bicubic | 32 | 29.54 | 0.8516 | 27.04 | 0.7551 | 26.78 | 0.7187 | 24.00 | 0.7144 | 26.16 | 0.8384 |
| DBDC+Pac (Tu et al., 2023) | 4 | 33.42 | 0.9143 | 29.69 | 0.8261 | 28.51 | 0.7869 | 27.05 | 0.8217 | 31.89 | 0.9274 |
| PTQ4ViT (Yuan et al., 2022) | 4 | 33.77 | 0.9201 | 29.75 | 0.8272 | 28.62 | 0.7942 | 27.43 | 0.8361 | 32.50 | 0.9360 |
| RepQ (Li et al., 2023) | 4 | 34.08 | 0.9232 | 30.04 | 0.8345 | 28.88 | 0.8013 | 27.87 | 0.8462 | 32.98 | 0.9401 |
| NoisyQuant (Liu et al., 2023) | 4 | 33.13 | 0.9122 | 29.06 | 0.8093 | 27.93 | 0.7754 | 26.66 | 0.8143 | 31.94 | 0.9293 |
| 2DQuant (Liu et al., 2024) | 4 | 34.06 | 0.9231 | 30.12 | 0.8374 | 28.89 | 0.7988 | 27.69 | 0.8405 | 32.88 | 0.9389 |
| CondiQuant (ours) | 4 | 34.32 | 0.9260 | 30.29 | 0.8417 | 29.05 | 0.8039 | 28.05 | 0.8506 | 33.23 | 0.9431 |
| DBDC+Pac (Tu et al., 2023) | 3 | 30.91 | 0.8445 | 28.02 | 0.7538 | 26.99 | 0.6937 | 25.10 | 0.7122 | 28.84 | 0.8403 |
| PTQ4ViT (Yuan et al., 2022) | 3 | 32.75 | 0.9028 | 29.14 | 0.8113 | 28.06 | 0.7712 | 26.43 | 0.8014 | 31.20 | 0.9131 |
| RepQ (Li et al., 2023) | 3 | 31.04 | 0.8548 | 28.04 | 0.7572 | 26.83 | 0.7019 | 25.56 | 0.7493 | 30.16 | 0.8904 |
| NoisyQuant (Liu et al., 2023) | 3 | 30.78 | 0.8511 | 27.94 | 0.7624 | 26.98 | 0.7153 | 25.43 | 0.7481 | 29.64 | 0.8792 |
| 2DQuant (Liu et al., 2024) | 3 | 33.24 | 0.9135 | 29.56 | 0.8255 | 28.50 | 0.7873 | 26.65 | 0.8116 | 31.46 | 0.9235 |
| CondiQuant (ours) | 3 | 33.92 | 0.9224 | 30.02 | 0.8367 | 28.84 | 0.7986 | 27.37 | 0.8356 | 32.48 | 0.9367 |
| DBDC+Pac (Tu et al., 2023) | 2 | 29.96 | 0.8254 | 27.53 | 0.7507 | 27.05 | 0.7136 | 24.57 | 0.7117 | 27.23 | 0.8213 |
| PTQ4ViT (Yuan et al., 2022) | 2 | 29.96 | 0.7901 | 27.36 | 0.7030 | 26.74 | 0.6590 | 24.56 | 0.6460 | 27.37 | 0.7390 |
| RepQ (Li et al., 2023) | 2 | 27.32 | 0.6478 | 25.63 | 0.5918 | 25.44 | 0.5652 | 23.42 | 0.5582 | 24.51 | 0.5721 |
| NoisyQuant (Liu et al., 2023) | 2 | 27.53 | 0.6641 | 25.77 | 0.5952 | 25.37 | 0.5613 | 23.59 | 0.5739 | 26.03 | 0.6632 |
| 2DQuant (Liu et al., 2024) | 2 | 31.62 | 0.8887 | 28.54 | 0.8038 | 27.85 | 0.7679 | 25.30 | 0.7685 | 28.46 | 0.8814 |
| CondiQuant (ours) | 2 | 33.00 | 0.9130 | 29.44 | 0.8253 | 28.45 | 0.7882 | 26.36 | 0.8080 | 30.88 | 0.9203 |

| Method | Bit | Set5 (×4) | | Set14 (×4) | | B100 (×4) | | Urban100 (×4) | | Manga109 (×4) | |
|---|---|---|---|---|---|---|---|---|---|---|---|
| | | PSNR↑ | SSIM↑ | PSNR↑ | SSIM↑ | PSNR↑ | SSIM↑ | PSNR↑ | SSIM↑ | PSNR↑ | SSIM↑ |
| SwinIR-light (Liang et al., 2021) | 32 | 32.45 | 0.8976 | 28.77 | 0.7858 | 27.69 | 0.7406 | 26.48 | 0.7980 | 30.92 | 0.9150 |
| Bicubic | 32 | 27.56 | 0.7896 | 25.51 | 0.6820 | 25.54 | 0.6466 | 22.68 | 0.6352 | 24.19 | 0.7670 |
| DBDC+Pac (Tu et al., 2023) | 4 | 30.74 | 0.8609 | 27.66 | 0.7526 | 26.97 | 0.7104 | 24.94 | 0.7369 | 28.52 | 0.8697 |
| PTQ4ViT (Yuan et al., 2022) | 4 | 31.49 | 0.8831 | 28.04 | 0.7680 | 27.20 | 0.7240 | 25.53 | 0.7660 | 29.52 | 0.8940 |
| RepQ (Li et al., 2023) | 4 | 31.77 | 0.8880 | 28.32 | 0.7750 | 27.40 | 0.7310 | 25.83 | 0.7780 | 29.88 | 0.9010 |
| NoisyQuant (Liu et al., 2023) | 4 | 31.09 | 0.8751 | 27.75 | 0.7591 | 26.91 | 0.7151 | 25.07 | 0.7500 | 28.96 | 0.8820 |
| 2DQuant (Liu et al., 2024) | 4 | 31.77 | 0.8867 | 28.30 | 0.7733 | 27.37 | 0.7278 | 25.71 | 0.7712 | 29.71 | 0.8972 |
| CondiQuant (ours) | 4 | 32.09 | 0.8923 | 28.50 | 0.7792 | 27.52 | 0.7345 | 25.97 | 0.7831 | 30.16 | 0.9054 |
| DBDC+Pac (Tu et al., 2023) | 3 | 27.91 | 0.7250 | 25.86 | 0.6451 | 25.65 | 0.6239 | 23.45 | 0.6249 | 26.03 | 0.7321 |
| PTQ4ViT (Yuan et al., 2022) | 3 | 29.77 | 0.8337 | 27.00 | 0.7248 | 26.21 | 0.6735 | 24.22 | 0.6983 | 27.94 | 0.8479 |
| RepQ (Li et al., 2023) | 3 | 27.52 | 0.7419 | 24.84 | 0.5996 | 23.99 | 0.5351 | 22.42 | 0.5739 | 26.58 | 0.7838 |
| NoisyQuant (Liu et al., 2023) | 3 | 28.90 | 0.7972 | 26.50 | 0.6970 | 26.16 | 0.6628 | 23.86 | 0.6667 | 27.17 | 0.8116 |
| 2DQuant (Liu et al., 2024) | 3 | 30.90 | 0.8704 | 27.75 | 0.7571 | 26.99 | 0.7126 | 24.85 | 0.7355 | 28.21 | 0.8683 |
| CondiQuant (ours) | 3 | 31.62 | 0.8855 | 28.20 | 0.7715 | 27.31 | 0.7269 | 25.39 | 0.7624 | 29.29 | 0.8915 |
| DBDC+Pac (Tu et al., 2023) | 2 | 25.01 | 0.5554 | 23.82 | 0.4995 | 23.64 | 0.4544 | 21.84 | 0.4631 | 23.63 | 0.5854 |
| PTQ4ViT (Yuan et al., 2022) | 2 | 27.23 | 0.6702 | 25.38 | 0.5914 | 25.15 | 0.5621 | 22.94 | 0.5587 | 24.66 | 0.6132 |
| RepQ (Li et al., 2023) | 2 | 25.55 | 0.5834 | 23.54 | 0.4751 | 23.30 | 0.4298 | 21.62 | 0.4493 | 23.60 | 0.5561 |
| NoisyQuant (Liu et al., 2023) | 2 | 25.94 | 0.5862 | 24.33 | 0.5067 | 24.16 | 0.4718 | 22.32 | 0.4841 | 23.82 | 0.5403 |
| 2DQuant (Liu et al., 2024) | 2 | 29.53 | 0.8372 | 26.86 | 0.7322 | 26.46 | 0.6927 | 23.84 | 0.6912 | 26.07 | 0.8163 |
| CondiQuant (ours) | 2 | 30.64 | 0.8671 | 27.59 | 0.7567 | 26.93 | 0.7136 | 24.54 | 0.7282 | 27.67 | 0.8613 |

## 4.3 COMPARISON WITH STATE-OF-THE-ART METHODS

We adopt two kinds of SOTA PTQ methods for comparison. The first kind is PTQ methods specifically for SR, including DBDC+Pac (Tu et al., 2023), 2DQuant (Liu et al., 2024). The second kind includes PTQ4ViT (Yuan et al., 2022), RepQ (Li et al., 2023), and NoisyQuant (Liu et al., 2023), which are designed for general vision transformers.

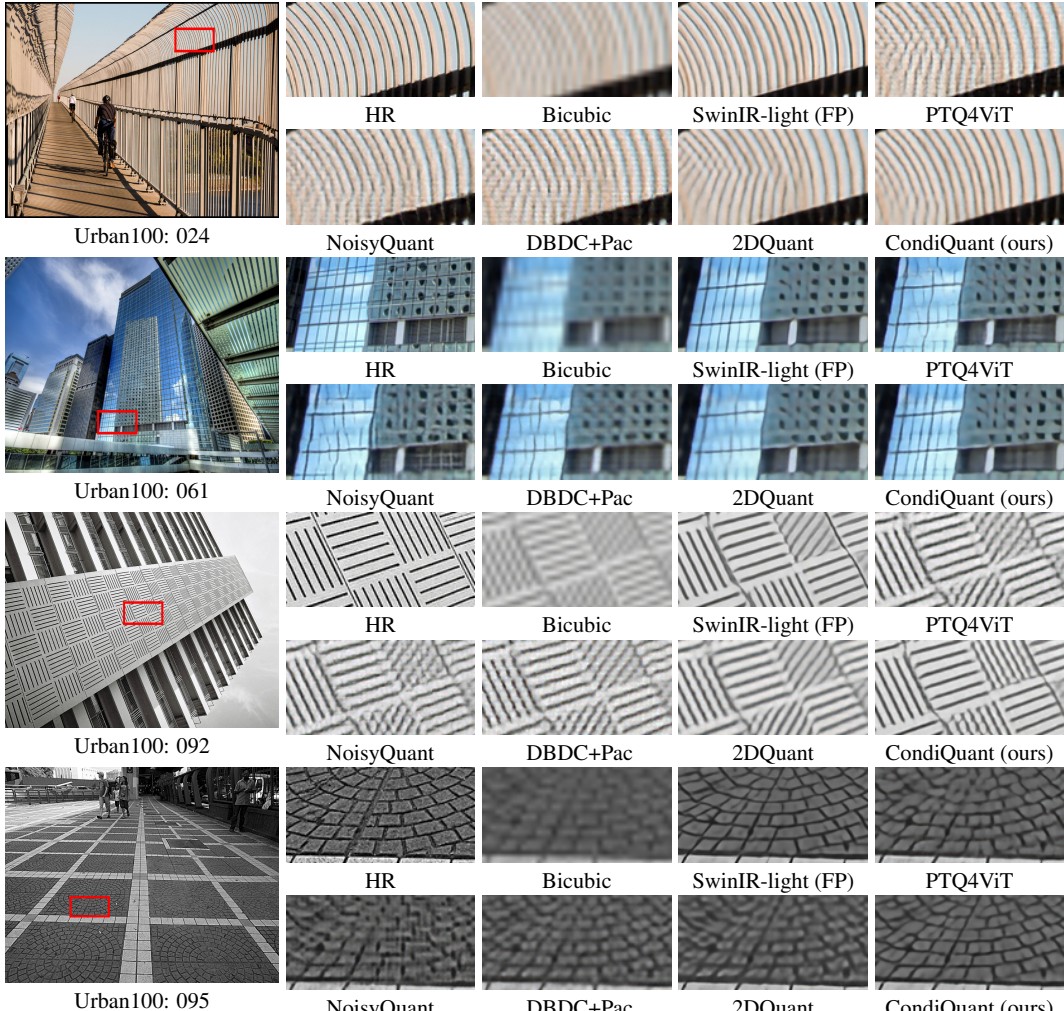

Figure 4: Visual comparison for image SR. We compare our proposed CondiQuant with current competitive quantization methods and the full-precision (FP) model. The visual results illustrate that CondiQuant gains sharper edges and reasonable textures.

**Quantitative Results.** Table 3 shows the comprehensive results comparing with edge PTQ methods with $2 \sim 4$ bits and $2 \sim 4$ scale factors on five common SR benchmarks. With two universally accepted metrics, our proposed CondiQuant outperforms all other methods on all benchmarks regardless of bit-width and scale factor. Specifically, CondiQuant has a huge improvement of 1.11 dB and 1.60 dB on $\times 4$, 2-bit on Set5 and Manga109 datasets, respectively. Besides, the average difference between $\times 2$, 4 bits CondiQuant, and the full-precision model is 0.38 dB. This means the quantization degradation when compressing to only 4 bits is minimal and small enough for real-world deployment on edge devices.

With condition number optimization, the model's sensitivity to the quantization error is reduced. Thereafter, the weight matrix is efficiently converted into a more quantization-friendly representation. Furthermore, the quantized model enjoys significant improvement during the following calibration and distillation stage. Besides, the oscillation during distillation is reduced, and detailed supporting information is in the supplemental material.

**Visual Results.** We present the visual comparison results of $\times 4$ in Fig. 4. The competing methods are struggling to recover inerratic textures and often generate artifacts. On the contrary, with our proposed CondiQuant, the quantized model could reconstruct HR with rich details and reasonable structures. In 024 and 061, the direction of the texture is distorted, and jagged edges are generated with Bicubic. Given the misleading input, other methods can not provide robust restoration like the FP model. However, our proposed CondiQuant can still guarantee fidelity to the greatest extent.

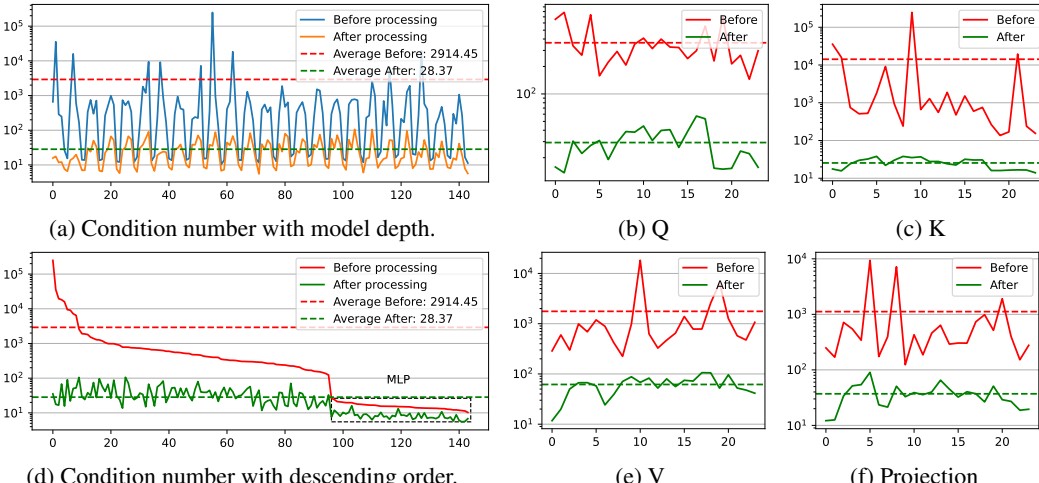

(a) Condition number with model depth.      (b) Q      (c) K

(d) Condition number with descending order.      (e) V      (f) Projection

Figure 5: Condition number before and after CondiQuant on ×2 model. Before CondiQuant, the distribution of condition numbers is extremely high, with an average of nearly 3k. After CondiQuant, the average value of condition numbers is significantly reduced to 28.37.

Even more, in 092, the FP model can not provide results with accurate direction on the wall, while CondiQuant can correct the error and provide sharp contents. This is because the FP model suffers from over-fitting, and CondiQuant could ease this phenomenon. Besides, in 095, the competing methods recover bricks with vague texture while CondiQuant remains aligned with the FP model. In conclusion, compared to other methods, CondiQuant achieves the minimum information loss after low bit quantization and restores images with high-fidelity, rich details, and sharper edges.

### 4.4 Condition Number Analysis

The excellent results are closely related to the decrease in condition number. Therefore, we visualize the condition number shifts in Fig. 5 on the SR model with the scale factor of 2. Figures 5a and 5d show the condition number of all layers (including Q, K, V, projection, FC1, and FC2) with different sequences. Fig. 5a is about model depth, and Fig. 5d sorts the condition number before CondiQuant with descending order. The other four figures show different layers with model depth as the X-axis.

Overall, the average condition number is hugely reduced from nearly 3k to merely 28.37. This evident decrease shows the effectiveness of the proximal descent step in CondiQuant. To be specific, a regular pattern is observed with model depth. The condition numbers of the FC1 and FC2 in the MLP layer are distinctly smaller than those of other layers, and their influence is also minor. Hence, considering efficiency, we do not perform CondiQuant on FC1 and FC2. The condition number of the K matrix in self-attention is usually extremely large. This is also consistent with the claim that the quantization degradation of ViT is mainly attributed to self-attention. Moreover, the distribution of the condition number after CondiQuant is greater in the middle and smaller in both ends. This indicates that the beginning and end modules are more important and sensitive in image restoration. To conclude, our proposed CondiQuant significantly reduces the condition number with high efficiency.

## 5 Conclusion

In this paper, we propose CondiQuant, a condition number based post-training quantization method for image super-resolution. We analyze that the degradation of quantization is attributed to the quantization of activation and build its relationship with condition number. Thereafter, we formulate the optimization problem to minimize the condition number while maintaining the output as it is. To optimize, we design the gradient descent step to keep the output still and the proximal descent step to reduce the condition number. Both steps are calculation-efficient and the entire iteration process takes only 19.0 seconds. As there's no additional module, we reach the theoretically optimal compression and speedup ratio. Specifically, when quantized to 2 bits, the compression ratio is $3.60\times$ and the speedup ratio is $5.08\times$. The comparison experiments demonstrate the excellent performance of CondiQuant while the ablation studies present its robustness.

## A    ETHICS STATEMENT

The research conducted in the paper conforms, in every respect, with the ICLR Code of Ethics.

## B    REPRODUCIBILITY STATEMENT

We have provided implementation details in Sec. 4. We will also release all the code and models.

## C    LLM USAGE STATEMENT

Large Language Models (LLMs) were used solely for polishing writing. They did not contribute to the research content or scientific findings of this work.

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
