# CONDIQUANT: CONDITION NUMBER BASED LOW-BIT QUANTIZATION FOR IMAGE SUPER-RESOLUTION

## 1 DERIVATION OF PROXIMAL GRADIENT DESCENT

In convex optimization problems, for differentiable objective functions, we can iteratively find the optimal solutions simply using gradient descent. For non-differentiable objective functions, we can still iterate toward the optimal solution by introducing subgradients. However, subgradient methods are much slower compared to gradient descent ones. To address this, for functions that are non-differentiable in their entirety but can be separated into differentiable and non-differentiable components, we can use a faster algorithm known as the proximal gradient descent method. Generally, consider an objective function that can be decomposed into two functions of the following forms:

$$f(x) = g(x) + h(x), \tag{1}$$

where $g(x)$ is a convex and differentiable function, while $h(x)$ is also convex but non-differentiable function. Using proximal gradient descent, a convergence rate of $O(1/\epsilon)$ can be achieved, where $\epsilon = f(x^k) - f(x^*)$ represents the deviation between the current iteration result and the optimal solution. Moreover, by accelerating the proximal gradient, a convergence rate of $O(1/\sqrt{\epsilon})$ can be attained.

**General form.** For the differentiable part $g(x)$, assuming the starting point is at $x_0$, the second-order Taylor expansion can be expressed as:

$$g(x) \approx g(x_0) + \nabla g(x_0)^T (x - x_0) + \frac{1}{2} \nabla^2 g(x_0)(x - x_0)^2. \tag{2}$$

Assuming $g(x)$ satisfies $L-$Lipschitz continuity, where the upper bound of the Hessian matrix is $L$. By setting the step size $\eta = \frac{1}{L}$, the second-order term can be simplified to $\frac{1}{2\eta}$. The problem can be written as:

$$\min_x \frac{1}{2\eta}[(x - x_0)^2 + 2\eta \nabla g(x_0)(x - x_0) + (\eta \nabla g(x_0))^2]$$
$$+ h(x)$$
$$= \min_x \frac{1}{2\eta}[x - x_0 + t\nabla g(x_0)]^2 + h(x)$$
$$= \min_x \frac{1}{2\eta}\|x - (x_0 - t\nabla g(x_0))\|_2^2 + h(x)$$
$$= \min_x \frac{1}{2\eta}\|x - z\|_2^2 + h(x),$$

where $z = x_0 - t\nabla g(x_0)$ is the form of gradient descent for $g(x)$. Therefore, its solution is:

$$x = \arg\min_x \frac{1}{2}\|x - z\|_2^2 + \eta g(x), \tag{3}$$

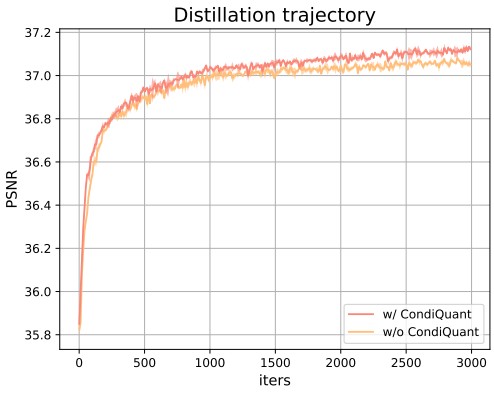 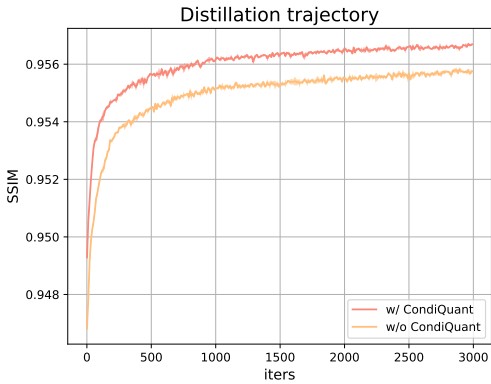

(a) Comparison of distillation trajectories of PSNR on Set5.

(b) Comparison of distillation trajectories of SSIM on Set5.

Figure 1: Condition number before and after CondiQuant on $\times 2$ model. Before CondiQuant, the distribution of condition numbers is extremely high, with an average of nearly 3k. After CondiQuant, the average value of condition numbers is significantly reduced to 28.37.

Eq. 3 is also known as the proximal operator, denoted as:

$$\text{prox}_{\eta,g}(z) = \arg \min_x \frac{1}{2} \|x - z\|_2^2 + \eta \cdot h(x). \tag{4}$$

Using Proximal Functions, we can define an iterative process for optimization, called the proximal descent method. First, choose an initial point $x_0$ and then iterate:

$$x_i = \text{prox}_{h,\eta} \left( x_{i-1} - \eta_i \nabla g(x_{i-1}) \right), \quad i = 1, 2, 3, \ldots \tag{5}$$

There are some advantages of using this method:

1. For many $h$ functions, the proximal projection $\text{prox}_{h,\eta}$ has an analytical solution.

2. The $\text{prox}_{h,\eta}$ depends only on $h$, so it can be used for different functions $g$.

3. $g$ can be any convex function, as long as we can compute its gradient.

---

**Algorithm 1:** CondiQuant pipeline

**Input:** Pretrained weight matrix $W$; Activation $X$ from calibration set; Learning rate or step size $\eta$; Penalty parameter $\lambda$; Maximum iterations $K$.

**Output:** Quantization friendly weight matrix $W'$.

1 $U, \Sigma, V \leftarrow \text{SVD}(W)$;
2 $Y \leftarrow XW$;

3 **for** $k = 1$ **to** $K$ **do**
4     $W_{k+\frac{1}{2}} \leftarrow W_k - \eta \nabla_W \frac{1}{2} \|Y - XW\|_F^2 \Big|_{W=W_k}$;
5     $U_k, \Sigma_k, V_k \leftarrow \text{SVD}(W_k)$;
6     $t \leftarrow \text{mean}(W_k)$;
7     $\Sigma_k^* \leftarrow (\Sigma_k + 2\lambda\mu t)/(1 + 2\lambda\mu)$;
8     $W_{k+1} \leftarrow U_k \Sigma_k^* V_k$;
9     $k \leftarrow k + 1$;
10 **end for**
11 $W' \leftarrow W_K$;
12 **return** $W'$;

---

**Accelerated proximal gradient method.** Choose an initial point $x_{-1} = x_0$, and then repeat the following steps:

1. Use the result from the previous step as the momentum and calculate $z$:

$$z = x_{k-1} + \frac{k-2}{k+1}\left(x_{k-1} - x_{k-2}\right), \tag{6}$$

2. Update $x$ using $z$:

$$x_k = \text{prox}_{\eta_k}\left(z - \eta_k \nabla g(z)\right), \tag{7}$$

The first step of $k=1$ is the standard proximal gradient update. Afterward, we use $z$ to replace $x_{k-1}$, transferring momentum from previous iterations to update the next gradient direction, making it align more closely with the current gradient.

## 2    CALIBRATION DETAILS

**Implementation details.** We adopted a per-channel granularity and quantize all the linear layers within the transformer blocks. Although some layers are full-rank, we still perform CondiQuant, given their limited impact. We made some modifications to the DOBI(Distribution-oriented bound initialization) algorithm of 2DQuant Liu et al. (2024) simultaneously. Specifically, for the long-tail distribution activations, we narrow the search range to 10 evenly spaced numbers between the 90th percentile and the maximum value. For weights and activations that follow an approximately symmetric distribution, we will search both the upper and lower bounds. The upper bound search range is from the 90th percentile to the maximum value, while the lower bound range is from the minimum value to the 10th percentile. We will also search 10 values within each range.

**Visualization of distillation.** We provide more visual comparison results in knowledge distillation to demonstrate the role of CondiQuant. It is performed based on 2-bit quantization of SwinIR-light x4. We changed different random seeds for multiple distillations and obtained the distillation trajectory with a 95% confidence interval as shown in Fig. 1a and Fig. 1b.

## 3    MORE VISUAL RESULTS

We provide more visual comparison results in Figures 2 and 3. All results demonstrate the outstanding performance of our proposed CondiQuant.

---

**Algorithm 2:** Modified DOBI pipeline

---

**Input:** Data to be quantized $v$, bit $b$, number of search point $K$.
**Output:** Clip bound $l$, $u$
1  $l \leftarrow \min(v)$;
2  $u \leftarrow \max(v)$;
3  $min\_mse \leftarrow +\infty$;
4  **if** $v$ *is symmetrical* **then**
5     $v_{10} \leftarrow$ 10th percentile of $v$;
6     $\Delta l \leftarrow (v_{10} - \min(v))/K$;
7  **else**
8     $\Delta l \leftarrow 0$;
9  **end if**
10  $v_{90} \leftarrow$ 90th percentile of $v$;
11  $\Delta u \leftarrow (\max(v) - v_{90})/K$;
12  **for** $i = 1$ **to** $K$ **do**
13     $l_i \leftarrow l + i \times \Delta l$;
14     $u_i \leftarrow u - i \times \Delta u$;
15     get $v_q$ based on quantization formula.;
16     $mse \leftarrow \|v - v_q\|_2$;
17     **if** $mse \leq min\_mse$ **then**
18        $min\_mse \leftarrow mse$;
19        $l\_best \leftarrow l_i, u\_best \leftarrow u_i$;
20     **end if**
21  **end for**
22  **return** $l\_best, u\_best$;

---

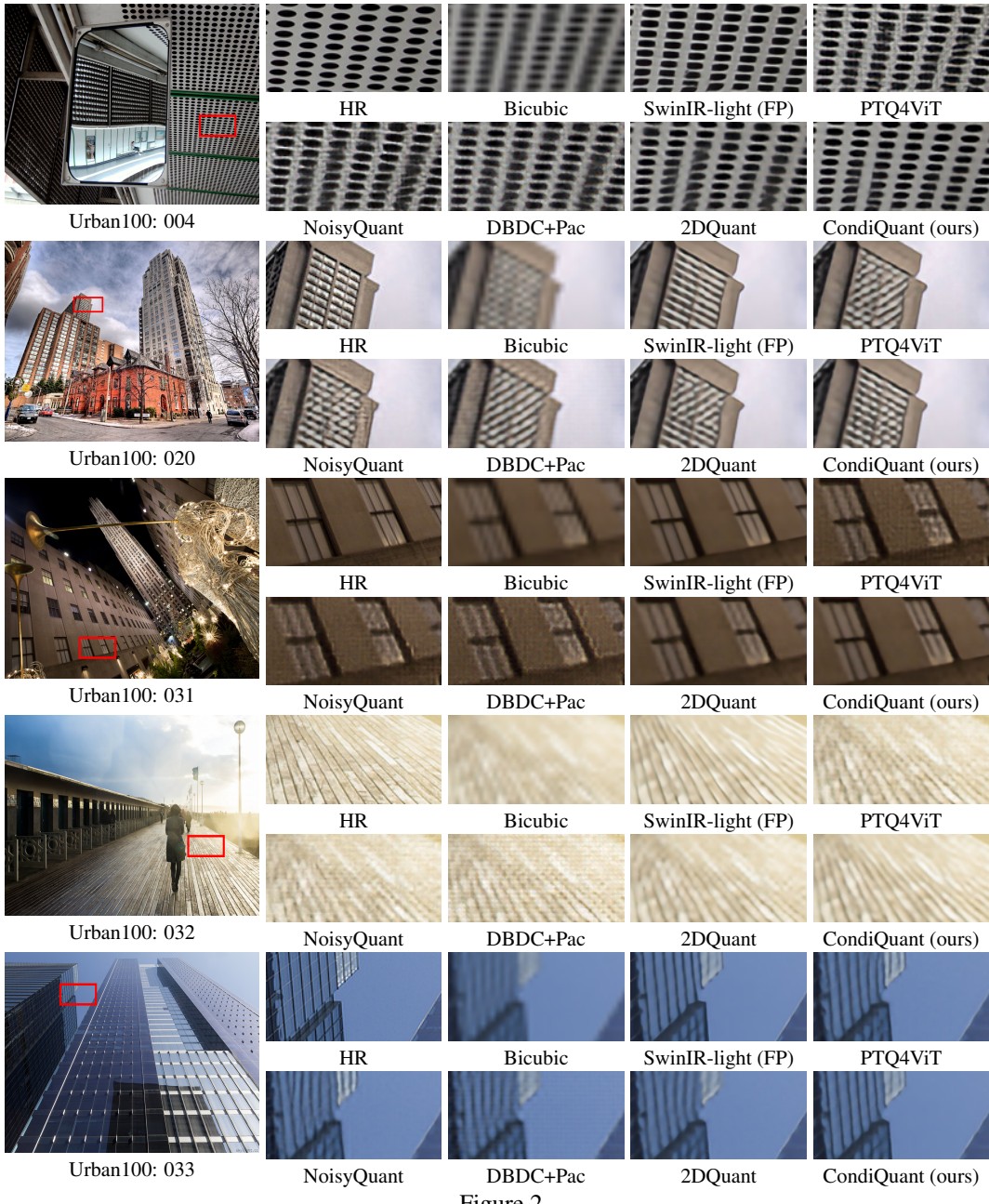

Figure 2

# REFERENCES

Kai Liu, Haotong Qin, Yong Guo, Xin Yuan, Linghe Kong, Guihai Chen, and Yulun Zhang. 2dquant: Low-bit post-training quantization for image super-resolution. *NeurIPS*, 2024.

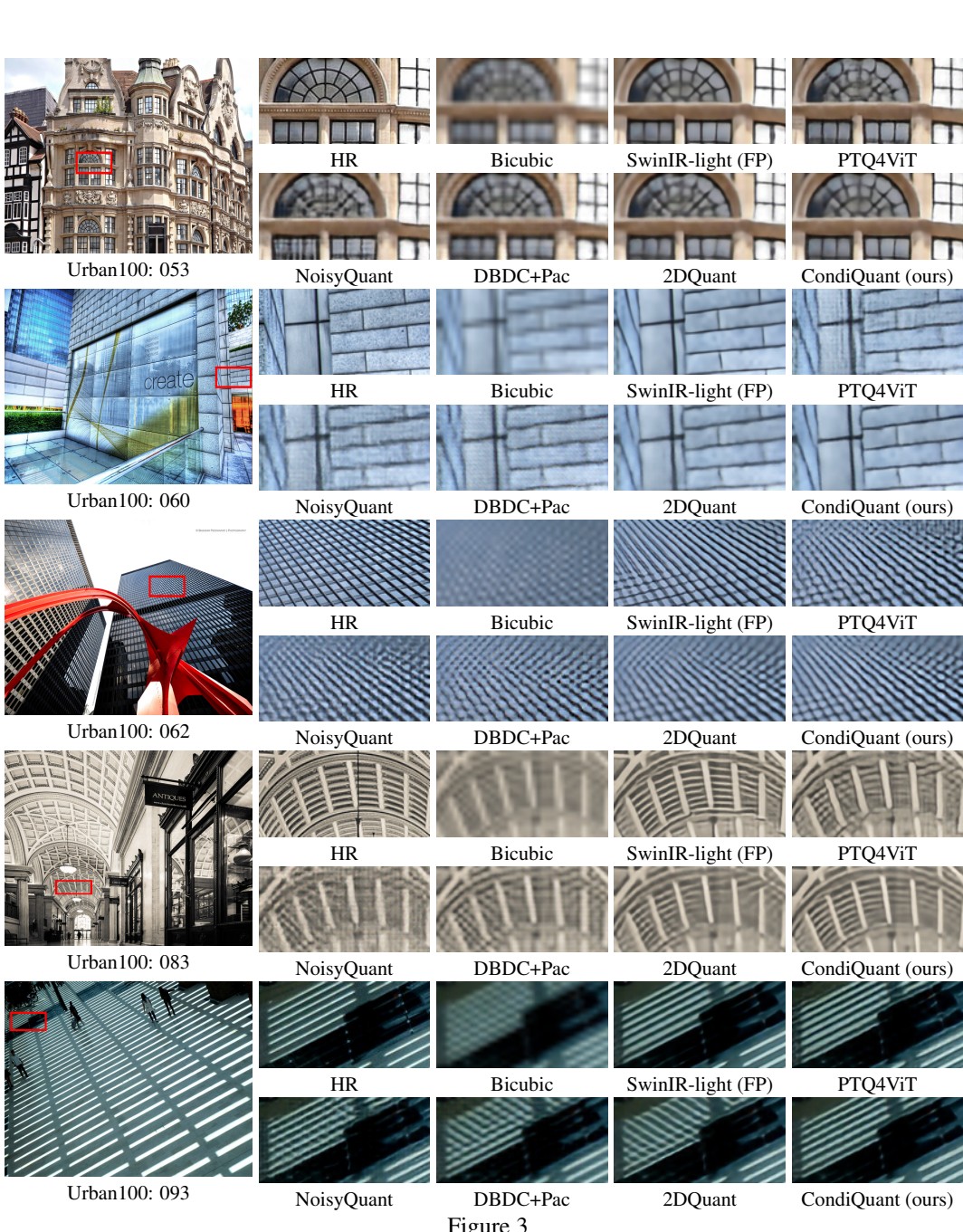

Figure 3