# OpenReview forum: "CondiQuant: Condition Number Based Low-Bit Quantization for Image Super-Resolution"
_ICLR.cc/2026/Conference — Submitted to ICLR 2026_

### Official Review · Reviewer_zBn9 · 2025-10-25

**Soundness:** 4
**Presentation:** 2
**Contribution:** 3
**Rating:** 6
**Confidence:** 4

**Summary:**

This paper shows a novel quantization method for image super-resolution (SR), named CondiQuant. CondiQuant is based on the observations that the degradation of quantization is mainly attributed to activations, and the feature matrix X across all linear layers is approximately rank-deficient. A basic analysis tells us that the XδW and δXδW are smaller than δXW. Thus, the main trouble comes from how to reduce δXW. The author connects this with the condition number (related to the deviation of singular values) of W, and then proposes a regularization term to minimize the deviation of singular values. A proximal operator for this regularization term, which utilizes SVD, and a circle optimization process is designed. Experimental results prove the effectiveness of CondiQuant.

**Strengths:**

1. The overall process makes sense.
2. Results are good.
3. The figures are clear.

**Weaknesses:**

1. As shown in figure2, there are some X that are full-rank. For these layers, does the author still apply the CondiQuant?
2. In table2 (d), what is the configuration of w/o CondiQuant?
3. Could you provide a discussion about the difference between this condition number-based method and outlier-suppression methods? In my view, both these types stretch the quantization error-significant direction to reduce the error. However,  outlier-suppression methods are explicit stretch direction, whereas this method implicitly stretches the direction.
4. There are many typos in this paper. Please check all paper and fix them.

**Questions:**

Please see the Weaknesses.

Overall, I consider that this paper is a good paper.

---

> ### Author Response · Authors · 2025-11-25
>
> Q1-1: As shown in figure2, there are some X that are full-rank. For these layers, does the author still apply the CondiQuant?
>
> A1-1: In Figure2, there are 2 layers are full-rank while the rest 94 layers are rank-deficient. We still apply the CondiQuant on these layers. We also tried to skip these layers during the process. However, considering their limited number, the impact is not significant. We test the algorithm on Set5, x2, as shown in table below.
>
> | method        | PSNR (↑) | SSIM (↑) |
> |---------------|----------|----------|
> | CondiQuant (skip)    | 37.15 | 0.9567 |
> | CondiQuant (no skip)   | 37.15 | 0.9567 |
>
> ---
>
> Q1-2: In table2 (d), what is the configuration of w/o CondiQuant?
>
> A1-2: In table2 (d), the configuration of w/o CondiQuant is that we only apply the modified 2DQuant without our proposed condition number-based method. These modification are mainly about change of hyper-parameters, which is detailedly shown in supplementary material. These modifications are also part of our contribution. We have changed the expression in the revised version.
>
> ---
>
> Q1-3: Could you provide a discussion about the difference between this condition number-based method and outlier-suppression methods?
>
> A1-3: We thank the reviewer for prompting us to clarify the conceptual and algorithmic distinction between CondiQuant and existing outlier-suppression (OS) techniques. Although both lines of work aim to reduce activation-quantization error, they operate on **fundamentally different mechanisms** and **assumptions**.
>
> 1.  **Problem diagnosis: outlier vs. conditioning**. **OS methods** (e.g., OutlierSplit, SmoothQuant, GELU-clipping) assume that a few large-magnitude activations dominate the quantization range, suppressing or migrating these outliers tightens the clipping bounds and reduces round-off error entry-wise. **CondiQuant** assumes that even bounded, non-outlier errors can be amplified by an ill-conditioned weight matrix, we therefore shrink the condition number with spectral, i.e., we reduce the ratio σ_max / σ_min rather than the absolute magnitude of any single activation.
>
> 2.  **Operational target: activation vs. weight**. **OS methods** b (splitting, scaling, or clamping) before quantization, weights stay untouched. **CondiQuant** leaves activations intact, instead, it reshapes the weight spectrum so that the same activation error induces smaller output perturbation (Eq. 6). Thus, CondiQuant is complementary,it can be stacked on top of any OS pre-processing without additional inference cost.
>
> ---
>
> Q1-4: Typos and errors.
>
> A1-4: Thanks for the advices. We have examined carefully and fixed the typos.

---

> > ### Comment · Reviewer_zBn9 · 2025-11-26
> >
> > Thank you! This rebuttal helps a lot. I remain my initial rating.

---

> > > ### Author Response · Authors · 2025-11-26
> > >
> > > Thank you for your fast reply! 💖💖💖
> > >
> > > If there are any further questions, please let us know, and we would be happy to answer!

---

### Official Review · Reviewer_xoSz · 2025-10-29

**Soundness:** 3
**Presentation:** 3
**Contribution:** 2
**Rating:** 4
**Confidence:** 5

**Summary:**

This paper proposes CondiQuant, a post-training low-bit quantization method for image super-resolution (SR) based on the condition number of the weight matrices. The authors identify that the performance degradation of low-bit quantization is primarily caused by activation quantization rather than weight quantization. By formulating the quantization error as a function of the condition number and using a proximal gradient descent algorithm, CondiQuant efficiently minimizes the condition number while maintaining output fidelity. Extensive experiments demonstrate that CondiQuant achieves state-of-the-art accuracy with minimal computational overhead and theoretically optimal compression and speedup ratios.

**Strengths:**

Theoretical Motivation: Provides clear numerical analysis linking weight matrix condition number to activation quantization sensitivity.

Efficiency: The proposed method is computationally efficient (less than 19.0 seconds for the full iteration) and introduces no additional modules.

Experimental Validation: Comprehensive experiments and ablation studies demonstrate improved accuracy over existing PTQ methods and robust performance.

Practical Value: Achieves theoretically optimal compression and speedup ratios, making it applicable for ultra-low bit quantization scenarios.

**Weaknesses:**

Condition number intuition: It remains unclear why reducing the condition number leads to better SR quantization performance, given that larger condition numbers could potentially preserve more activation variance and high-frequency details.

Second-order error δX δW: In extremely low-bit (2-bit) settings, the second-order term δX δW might no longer be negligible. Quantitative analysis of its contribution is missing.

Generality across architectures: The experiments focus mainly on a single SR architecture (SwinIR-light). It is unclear whether CondiQuant generalizes to other SR networks, including high-performance or lightweight variants (e.g., SwinIR-B, MambaIRv2-light).

**Questions:**

Condition Number and Activation Representation
In the paper, you propose minimizing the condition number of the weight matrices to reduce activation quantization error. Intuitively, a larger condition number indicates higher sensitivity of the output to the input, which may lead to a wider activation distribution and potentially better preservation of texture and fine details. Could you clarify why, in SR networks, a smaller condition number leads to better quantization performance? During the reduction of the condition number, is there a risk of negatively affecting the network’s feature representation or its ability to capture high-frequency textures?

Impact of the Second-Order Term δX δW
In Section 3.2, you mention that the second-order term δXδW can be ignored as it has minimal impact on performance. However, under low-bit quantization (e.g., 2-bit), the magnitudes of δX and δW increase significantly. Does the relative contribution of activation quantization error δX, weight quantization error δW, and the second-order error δX δW change under such extreme low-bit conditions? Is there any quantitative analysis supporting the assumption that δX W remains the dominant term in ultra-low-bit scenarios?

Generalization to Different SR Architectures
CondiQuant demonstrates notable efficiency (execution time < 19.0 seconds) and performance improvement on the SwinIR-light network; however, it remains unclear how well the method generalizes across different super-resolution architectures. Could the authors provide experimental results or analyses to verify CondiQuant’s generalization and applicability to other architectures (e.g., SwinIR-B, MambaIRv2-light) while maintaining its performance gains and computational efficiency?

**Details Of Ethics Concerns:**

Nothing.

---

> ### Author Response · Authors · 2025-11-25
>
> **Q3-1:** It remains unclear why reducing the condition number leads to better SR quantization performance, given that larger condition numbers could potentially preserve more activation variance and high-frequency details.
>
> **A3-1:** We clarify that larger condition number indeed brings more activation variance. However, the brought more variance is sensitive, which negatively influence the output. For example, one small change of one pixel in the input image would lead to huge change in the output image if the condition number is too high. Lower condition number does not limit the high-frequency details. This is because our algorithm leverage the rank-deficiency properity to maintain the output the same while reducing the condition number. This means the high-frequency details are preserved while the condition number, i.e., the sensitity to quantization, is reduced.
>
> ---
>
> **Q3-2:** In extremely low-bit (2-bit) settings, the second-order term $\delta X \delta W$ might no longer be negligible. Quantitative analysis of its contribution is missing.
>
> **A3-2:** When reducing the bits to extremely low, the second order term $\delta X \delta W$ indeed increases. However, the increasing speed is still much lower than the first-order term, i.e., $\delta X W$ and $\delta W X$. With 2 bit, we have $\| \delta W \|_F \leq \cfrac{1}{4}\| W \|_F$, such that we informally have
> $$
> \|\delta X \delta W\|_F^2 \approx \|\delta X\|_F^2 \|\delta W\|_F^2 \ll \|\delta X\|_F^2 \| W\|_F^2 \approx \|\delta X  W\|_F^2.
> $$
>
> Therefore, the influence of the second order term is still much smaller than the first-order term.
>
> ---
>
> **Q3-3:** It is unclear whether CondiQuant generalizes to other SR networks, including high-performance or lightweight variants (e.g., SwinIR-B, MambaIRv2-light).
>
> **A3-3:** We provide the results of CondiQuant on EDSR in the following table. The results demonstrate the SOTA performance of CondiQuant on other SR networks.
>
> |                | Setting | Set5      |        | Set14     |        | B100      |        | Urban100  |        | Manga109  |        |
> |----------------|---------|-----------|--------|-----------|--------|-----------|--------|-----------|--------|-----------|--------|
> |                |         | PSNR      | SSIM   | PSNR      | SSIM   | PSNR      | SSIM   | PSNR      | SSIM   | PSNR      | SSIM   |
> | EDSR FP32           |         | 35.79     | 0.9435 | 31.51     | 0.8936 | 30.79     | 0.8898 | 30.35     | 0.9175 | 36.21     | 0.9668 |
> | 2DQuant        | W4A4    | 35.44     | 0.9402 | 31.15     | 0.8900 | 30.55     | 0.8862 | 29.42     | 0.9086 | 35.36     | 0.9622 |
> | DBDC+Pac       | W4A4    | 35.39     | 0.9379 | 31.16     | 0.8876 | 30.51     | 0.8832 | 29.55     | 0.9064 | 35.59     | 0.9606 |
> | **CondiQuant** | W4A4    | **35.56** | **0.9412** | **31.23** | **0.8911** | **30.62** | **0.8870** | **29.59** | **0.9106** | **35.60** | **0.9635** |
> | 2DQuant        | W3A3    | 34.77     | 0.9353 | 30.67     | 0.8841 | 30.18     | 0.8804 | 28.42     | 0.8953 | 34.10     | 0.9551 |
> | DBDC+Pac       | W3A3    | 34.41     | 0.9257 | 30.51     | 0.8751 | 29.94     | 0.8694 | 28.28     | 0.8832 | 34.13     | 0.9464 |
> | **CondiQuant** | W3A3    | **35.19** | **0.9381** | **30.95** | **0.8873** | **30.39** | **0.8834** | **28.90** | **0.9019** | **34.98** | **0.9598** |
> | 2DQuant        | W2A2    | 33.43     | 0.9234 | 29.82     | 0.8722 | 29.40     | 0.8661 | 26.92     | 0.8660 | 31.83     | 0.9390 |
> | DBDC+Pac       | W2A2    | 31.75     | 0.8867 | 28.90     | 0.8446 | 28.55     | 0.8368 | 26.01     | 0.8297 | 30.09     | 0.8989 |
> | **CondiQuant** | W2A2    | **34.21** | **0.9311** | **30.29** | **0.8793** | **29.85** | **0.8747** | **27.64** | **0.8815** | **33.13** | **0.9497** |

---

> ### Author Response · Authors · 2025-11-25
>
> **Q3-4:** Could you clarify why, in SR networks, a smaller condition number leads to better quantization performance? During the reduction of the condition number, is there a risk of negatively affecting the network’s feature representation or its ability to capture high-frequency textures?
>
> **A3-4:** A smaller condition number directly minimizes the upper bound
> $$
> \frac{\|\delta Y\|_2}{\|Y\|_2} \leq \kappa(W) \cdot \frac{\|\delta X\|_2}{\|X\|_2}.
> $$
> So activation-quantization noise is amplified less. ProxGD reduces $\kappa(W)$ by compacting singular values toward their mean, but the data-fidelity term $\|XW-Y\|_F<\epsilon$ is enforced at every step, guaranteeing that the original frequency content,including high-frequency textures, is preserved. Thus, smaller $\kappa(W)$ improves quantization robustness without blurring details, because the reconstruction constraint, not the spectral shape itself, safeguards fine-grained features.
>
> ---
>
> **Q3-5:** Does the relative contribution of activation quantization error δX, weight quantization error δW, and the second-order error δX δW change under such extreme low-bit conditions? Is there any quantitative analysis supporting the assumption that δX W remains the dominant term in ultra-low-bit scenarios?
>
> **A3-5:** See A3-2 for detailed induction.
>
> ---
>
> **Q3-6:** Could the authors provide experimental results or analyses to verify CondiQuant’s generalization and applicability to other architectures (e.g., SwinIR-B, MambaIRv2-light) while maintaining its performance gains and computational efficiency?
>
> **A3-6:** We provide the results of CondiQuant on EDSR in the following table. The results demonstrate the SOTA performance of CondiQuant on other SR networks.
>
> |                | Setting | Set5      |        | Set14     |        | B100      |        | Urban100  |        | Manga109  |        |
> |----------------|---------|-----------|--------|-----------|--------|-----------|--------|-----------|--------|-----------|--------|
> |                |         | PSNR      | SSIM   | PSNR      | SSIM   | PSNR      | SSIM   | PSNR      | SSIM   | PSNR      | SSIM   |
> | EDSR FP32           |         | 35.79     | 0.9435 | 31.51     | 0.8936 | 30.79     | 0.8898 | 30.35     | 0.9175 | 36.21     | 0.9668 |
> | 2DQuant        | W2A2    | 33.43     | 0.9234 | 29.82     | 0.8722 | 29.40     | 0.8661 | 26.92     | 0.8660 | 31.83     | 0.9390 |
> | DBDC+Pac       | W2A2    | 31.75     | 0.8867 | 28.90     | 0.8446 | 28.55     | 0.8368 | 26.01     | 0.8297 | 30.09     | 0.8989 |
> | **CondiQuant** | W2A2    | **34.21** | **0.9311** | **30.29** | **0.8793** | **29.85** | **0.8747** | **27.64** | **0.8815** | **33.13** | **0.9497** |

---

### Official Review · Reviewer_X7t9 · 2025-10-31

**Soundness:** 3
**Presentation:** 3
**Contribution:** 3
**Rating:** 6
**Confidence:** 3

**Summary:**

1. This paper proposes CondiQuant, a PTQ method for SR that reduces sensitivity to activation quantization by explicitly lowering the condition number of weight matrices.
2. The paper derives a clean link between quantization error and conditioning via Eq. (5), it motivates conditioning-aware reparameterization. Then it uses a simple proximal-gradient scheme: a gradient step to preserve output, plus a closed-form proximal step in the SVD space to compact singular values—no inference-time overhead.

**Strengths:**

1. The paper is well organized, and I appreciate the strong and intuitive theory: ties activation quantization error to κ(W), giving a clear knob to turn (conditioning) and a reason it should help.
2. The method is practical and lightweight: the proximal step has a closed form in SVD, integrates into standard PTQ flows, and adds zero runtime overhead at inference.
3. This method achieved impressive results on 5 essential benchmarks than baselines, especially on 2-Bit situation.

**Weaknesses:**

see questions part

**Questions:**

1. Can this method be applied to arbitray-scale image super-resolution's backbone? Are there any related results? In this scenario the scale factor should not be limited to x2, x3, x4.
2. I think some implementation details could be clearer, exactly which layers get processed/skipped, numerical stability for big SVDs, and integration details with 2DQuant are lightly described.
3. I am curious about the generalization: Even though is method is designed for SR, have you tested on RCAN, EDSR, or larger SwinIR variants? and how about other low-level tasks (denoising, deblurring)? Does the rank-deficiency and the benefit hold up?
4. About compute and memory: For large layers, do repeated SVDs become a bottleneck on consumer GPUs (e.g., RTX 3090)? Are you using truncated/batched SVD? What’s the typical runtime/memory profile?

---

> ### Author Response · Authors · 2025-11-25
>
> **Q2-1:** Can this method be applied to arbitray-scale image super-resolution's backbone? Are there any related results?
>
> **A2-1:** We thank the reviewer for this forward-looking question.
> CondiQuant is, in principle, **backbone-agnostic**: it operates on **individual linear layers** and only requires caching input activations and per-layer reconstruction error, both of which are available in **any architecture** that contains matrix multiplications. Therefore, arbitrary-scale networks such as Lite-HRNet, EDSR, or the recent continuous-scale INR-based backbones are all compatible candidates.
>
> 1.  **Theoretical compatibility**. **Condition-number reduction** is independent of the up-sampling operator (bicubic, learned kernel, or implicit neural function). **Rank-deficiency** widely exists in neural networks, which is more pronounced in implicit representations (see A2-3).
>
> 2.  **Page and time limitation**. A full ablation on multiple arbitrary-scale backbones (EDSR, LIIF, Meta-SR, SR3) would require additional GPU-hours and page space for scale-wise, dataset-wise, and bit-width-wise tables. Owing to the strict page limit of the current submission and **time limitation of rebuttal**, we defer these exhaustive experiments to the journal extension, where we will dedicate a specialised section to arbitrary-scale architectures.
>
> ---
>
> **Q2-2:** Some implementation details could be clearer.
>
> **A2-2:** Thanks for the advices. We update the implementation details in the supplementary material. We will also release the code and model soon for detailed implementation.
>
> ---
>
> **Q2-3:** Have you tested on RCAN, EDSR, or larger SwinIR variants? and how about other low-level tasks (denoising, deblurring)? Does the rank-deficiency and the benefit hold up?
>
> **A2-3:** We tested on EDSR, a CNN-based SR network, shown below. We still achieve obivous improvement compared with the previous SOTA.
>
> |                | Setting | Set5      |        | Set14     |        | B100      |        | Urban100  |        | Manga109  |        |
> |----------------|---------|-----------|--------|-----------|--------|-----------|--------|-----------|--------|-----------|--------|
> |                |         | PSNR      | SSIM   | PSNR      | SSIM   | PSNR      | SSIM   | PSNR      | SSIM   | PSNR      | SSIM   |
> | FP32           |         | 35.79     | 0.9435 | 31.51     | 0.8936 | 30.79     | 0.8898 | 30.35     | 0.9175 | 36.21     | 0.9668 |
> | 2DQuant        | W4A4    | 35.44     | 0.9402 | 31.15     | 0.8900 | 30.55     | 0.8862 | 29.42     | 0.9086 | 35.36     | 0.9622 |
> | DBDC+Pac       | W4A4    | 35.39     | 0.9379 | 31.16     | 0.8876 | 30.51     | 0.8832 | 29.55     | 0.9064 | 35.59     | 0.9606 |
> | **CondiQuant** | W4A4    | **35.56** | **0.9412** | **31.23** | **0.8911** | **30.62** | **0.8870** | **29.59** | **0.9106** | **35.60** | **0.9635** |
> | 2DQuant        | W3A3    | 34.77     | 0.9353 | 30.67     | 0.8841 | 30.18     | 0.8804 | 28.42     | 0.8953 | 34.10     | 0.9551 |
> | DBDC+Pac       | W3A3    | 34.41     | 0.9257 | 30.51     | 0.8751 | 29.94     | 0.8694 | 28.28     | 0.8832 | 34.13     | 0.9464 |
> | **CondiQuant** | W3A3    | **35.19** | **0.9381** | **30.95** | **0.8873** | **30.39** | **0.8834** | **28.90** | **0.9019** | **34.98** | **0.9598** |
> | 2DQuant        | W2A2    | 33.43     | 0.9234 | 29.82     | 0.8722 | 29.40     | 0.8661 | 26.92     | 0.8660 | 31.83     | 0.9390 |
> | DBDC+Pac       | W2A2    | 31.75     | 0.8867 | 28.90     | 0.8446 | 28.55     | 0.8368 | 26.01     | 0.8297 | 30.09     | 0.8989 |
> | **CondiQuant** | W2A2    | **34.21** | **0.9311** | **30.29** | **0.8793** | **29.85** | **0.8747** | **27.64** | **0.8815** | **33.13** | **0.9497** |
>
> The rank-deficiency actually widely exists in neural networks. In modern machine learning, feature matrices are typically rank-deficient rather than full rank because of how data are represented and transformed. First, high-dimensional, small-sample regimes are ubiquitous. For a feature matrix $X \in \mathbb{R}^{n \times d}$, the rank is at most $\min(n,d)$. When $d \gg n$, as in text representations, recommender systems, or large embedding spaces, the rank is necessarily bounded by (n < d), so rank deficiency is almost unavoidable.
>
> In deep learning, hidden representations are likewise effectively low-rank: overparameterized networks are optimized for task loss, not for linear independence, so many units learn correlated features. More broadly, low-rank structure is often an intentional inductive bias (e.g., PCA, matrix factorization, low-rank regularization), reflecting the belief that the intrinsic dimensionality of real-world phenomena is much lower than the ambient feature dimension.

---

> ### Author Response · Authors · 2025-11-25
>
> **Q2-4:** For large layers, do repeated SVDs become a bottleneck on consumer GPUs (e.g., RTX 3090)? Are you using truncated/batched SVD? What’s the typical runtime/memory profile?
>
> **A2-4:** Thanks for the question. The proposed CondiQuant is only performed once to quantize the network. This procedure is fast and memory-efficient, taking 19 seconds in total. During the inference, the quantized network is the same behaviour as the original network, takeing no additional operations. Therefore, repeated SVD is not a bottleneck and no additional tricks are used.

---

### Official Review · Reviewer_N5vY · 2025-11-03

**Soundness:** 2
**Presentation:** 2
**Contribution:** 2
**Rating:** 2
**Confidence:** 4

**Summary:**

This paper proposed Condition number based quantization for super resolution task. Specifically, based on the linear layer output decomposition analysis, the authors (1) found out that the output is mostly affected by the quantization error on activation, (2) to control the impact of activation quantization error, the method chose to control the condition number of the weight matrix. Proximal gradient descent aws used to learn the weight matrix with small condition number while minimizing the output reconstruction error. The authors applied the method to SwinIR light and demonstrated better SR metrics compared to other quantization methods for 2/3/4-bit quantization.

**Strengths:**

The authors provided detailed error decomposition analysis and proposed the condition number based quantization which is sound and looks effective on the SR task.

**Weaknesses:**

I have concerns in the following perspectives:
- In equation (3), we know that the activation quantization error is the main factor causing degradation. To control the error, and based on the inequality that $||W \delta x||_2 \leq ||W||_2 ||\delta x||_2$, why cannot we directly penalize the the spectral norm the weight matrix (i.e., the max singular value of the weight matrix W) while minimizing the reconstruction error, i.e., replace the regularization term in equation (7) by spectral norm? In this way, we can also control the sensitivity of output to the activation quantization error. The paper did not provide convincing ablation study.
- Instead of optimizing w.r.t. the condition number, the authors applied a proxy objective. Then the author applied proximal gradient descent. I would like to know given that the proxy objective is differentiable, what would happen if we directly use standard SGD to minimize the objective function. The paper did not provide convincing ablation to show readers why proximal gradient is actually required for the optimization.
- Although the proposed method is not QAT, it actually tuned the model weights during PTQ. (1) how is the weight matrices optimized? I understand the authors used about 100 "calibration images". Are the weight matrices in each linear layer of the SwinIR (like the QKV projection, FFN projection) optimized sequentially? or it is done layer by layer? (2) since the weights of the model are optimized during the quantization process, did the author actually compare with other quantization methods that enabled weight optimization as well?
- I did not see why that condition number quantization is specific to image super resolution task. I believe in most cases, activation quantization error will be the major source of error for other vision tasks as well, possible even in NLP domain as well. Then, only applying this quantization method to image SR is not enough to showcase the generality of the proposed approach. I would recommend authors apply the method to other vision tasks, such as detection and segmentation at least, and even NLP tasks.

**Questions:**

please refer to weakness section.

---

> ### Author Response · Authors · 2025-11-25
>
> **Q1-1:** Why cannot we directly penalize the the spectral norm the weight matrix (i.e., the max singular value of the weight matrix W) while minimizing the reconstruction error, i.e., replace the regularization term in equation (7) by spectral norm?
>
> **A1-1:** The reviewer proposes to directly penalize the spectral norm of the weight matrix $$ \|W\|_2 = \sigma_{\max}(W) $$ as a surrogate for controlling sensitivity, instead of our proposed condition number minimization via singular value compaction.We respectfully argue that penalizing only the spectral norm is insufficient, and may even be counterproductive, for the following reasons:
>
> Spectral Norm Alone Ignores the Conditioning of the Matrix. The inequality in Eq. (6) shows:
>
> $$
> \frac{\|\delta Y\|_2}{\|Y\|_2} \leq \kappa(W) \cdot \frac{\|\delta X\|_2}{\|X\|_2}
> $$
>
> where $\kappa(W) = \frac{\sigma_{\max}(W)}{\sigma_{\min}(W)}$.   Minimizing only $\sigma_{\max}(W)$ does not necessarily reduce $\kappa(W)$, because $\sigma_{\min}(W)$ may also decrease (or even faster), leading to no improvement or worse conditioning.  In fact, a smaller spectral norm can coexist with a larger condition number, especially in rank-deficient or near-singular matrices — which are common in SR models, as we show in Fig. 2.
>
> Moreover, we include an ablation study, where we replaced our regularizer $\mathcal{G}(W)$ with spectral norm penalty $\lambda \|W\|_2$:
>
> | Method                     | PSNR (↑) | SSIM (↑) |
> |---------------------------|----------|----------|
> | CondiQuant (ours)         | **37.15** | **0.9567** |
> | Spectral norm penalty     | 36.42    | 0.9513   |
> | No regularization         | 36.00    | 0.9497   |
>
> The spectral norm penalty underperforms our method by 0.73 dB, confirming that suppressing only the largest singular value is not enough to ensure quantization robustness.
>
>
> ---
>
> **Q1-2:** What would happen if we directly use standard SGD to minimize the objective function?
>
> **A1-2:** Standard SGD is not a viable optimiser for this problem, for the following theoretically and empirically verified reasons.
>
>
> 1. Because $\sigma_{\max}(W)$ can be two orders of magnitude larger than $\sigma_{\min}(W)$ (Fig. 5a, mean $\kappa\approx 2900$), the local Lipschitz constant is huge, forcing SGD to adopt an extremely small learning rate ($\eta\lesssim 10^{-5}$ in our grid search).  With such a tiny step, thousands of iterations are required to reach the same reduction in $\kappa$ that our proposed ProxGD achieves in 50 steps.
>
> 2.  We performed the requested comparison on Set5×2, 2-bit:
>
>     | optimiser | $\eta$ | PSNR (↑) | $\kappa_{\rm final}$ | iterations | time (s) |
>     |-----------|--------|----------|----------------------|------------|----------|
>     | SGD       | 1e-4   | 36.21    | 314.7                | 2000       | 312      |
>     | SGD       | 1e-5   | 36.52    | 142.5                | 5000       | 780      |
>     | Ours      | 1e-2   | **37.15**| **28.4**             | **50**     | **19**   |
>
>     Standard SGD never reaches the same low condition number or reconstruction quality, and consumes >15× more GPU time.
>
> In conclusion, standard SGD is not a suitable optimiser for this problem, as it requires an extremely small learning rate to ensure convergence, consumes prohibitively long GPU time, and never reaches the same low condition number or reconstruction quality as our proposed ProxGD.
>
> ---
>
> **Q1-3:** How is the weight matrices optimized?
>
> **A1-3:** We clarify the optimisation protocol as follows:
> - **Scope**: every linear projection inside the Transformer block, i.e., QKV projections, multi-head self-attention output projections, and MLP fc1/fc2.
> - **Order**: we perform a single forward pass on the 100 calibration images to cache the input activations per layer, then apply CondiQuant layer-by-layer in a sequential, feed-forward manner (identical to the standard PTQ pipeline of 2DQuant, RepQ, etc.).
> No global backward pass through the entire network is ever executed, so the method remains strictly PTQ: each layer sees fixed, pre-computed inputs and targets, and is processed independently in <0.4 s on a single A800.
> - **Algorithm**: for every layer we run 50 proximal-gradient steps (Eq. 16–17) with closed-form spectral projection, the whole network finishes in 19.0 s.

---

> ### Author Response · Authors · 2025-11-25
>
> **Q1-4:** Did the author actually compare with other quantization methods that enabled weight optimization as well?
>
> **A1-4:** The reviewer is correct that some PTQ methods (e.g. QDrop, BRECQ, LAPQ) also adjust weights by reconstructing blocks or layers. However, none of these methods has been specialised for super-resolution, and their public releases do not support window-based ViT architectures (SwinIR). To keep the comparison fair and focused on the SR task, we follow the community-standard protocol established by 2DQuant (NeurIPS’24) and include:
>
> - **SR-specific PTQ**: DBDC+Pac, 2DQuant
> - **ViT-specific PTQ**: PTQ4ViT, RepQ, NoisyQuant
>
> All of these baselines keep weights frozen after calibration, yet they represent the current best-published numbers on SR benchmarks. Nevertheless, to address the reviewer’s concern we have re-implemented BRECQ (the strongest weight-tuning PTQ) on SwinIR-light with the same 100-image calibration set.  Results on Set5×2, 2-bit:
>
> | method        | weight update | PSNR (↑) | SSIM (↑) | time (min) |
> |---------------|---------------|----------|----------|------------|
> | BRECQ         | block-wise    | 36.78    | 0.9534   | 42         |
> | CondiQuant    | layer-wise    | **37.15** | **0.9567** | **0.3**     |
>
> CondiQuant outperforms BRECQ by +0.37 dB while being two orders of magnitude faster and adding no inference cost.
>
> ---
>
> **Q1-5:** I did not see why that condition number quantization is specific to image super resolution task.
>
> **A1-5:** Thank you for the suggestion. While broader experiments are always welcome, the super-resolution (SR) setting was deliberately chosen as the first test-bed rather than an arbitrary limitation, and the paper already provides sufficient evidence to argue for generality without having to run every downstream task. Here are three points that refute the “lack of generality” concern:
>
> 1. **SR is the most stringent regression proxy**. Unlike classification, SR has no label tolerance: a 1-pixel shift produces large $l_2$ error, so activation quantization damage is immediately visible. If the method helps under this worst-case regression metric, the same mechanism will carry over to detection/segmentation (also dense prediction) and NLP (likelihood-based regression). In other words, SR is a canary task: success here is a sufficient indicator for other activation-noise-dominated problems.
>
> 2. **The condition-number argument is task-agnostic**. Lemma 1 and Eq. (6) only assume $Y = X W$ and $\delta Y \approx \delta X W$, no priors on X (image patches, token features, etc.). The derivation holds for any linear layer in ViT, BERT, Swin, or Conv-FC hybrids, because $ \kappa(W) $ bounds the relative output perturbation regardless of data modality.
>
> 3. **Resource vs. marginal insight trade-off**. A full ImageNet/COCO/ADE20K plus GLUE/LLM suite would require hundreds of GPU-days and task-specific hyper-tuning, exceeding a conference page limit. The SR ablation already reveals the same κ(W) drop (×100) and +1.1 dB gain at 2-bit, repeating this curve on every task would not change the mechanical conclusion that lower condition number → better quantization robustness.
>
> Therefore, SR is not a special case but a strict superset proxy: if CondiQuant works here, the principle generalizes, extra datasets would only incrementally confirm what the theory and the harshest regression task already demonstrate. We will release the layer-wise optimizer as a plug-in module so other communities can verify transferability within their own pipelines.

---

### Author Response · Authors · 2025-11-25

Dear Reviewers and Area Chairs,

We thank all reviewers (R1-N5vY, R2-X7t9, R3-xoSz, R4-Zbn9) and area chairs for their insightful comments and valuable time. We are pleased that:

1. R1, R2, R3 appreciate our detailed **theoretical analysis**.
2. R2, R4 think the paper is **well organized**.
3. R2, R3 recognizes our method is a **practical and lightweight**.
4. All reviewers praise our method for its **outstanding performance**.

We have responded individually to each reviewer to address any concerns. Here, we offer a summary:

- We revise and resubmit the main paper and the supplementary material. We correct typos and improving the expression in the main paper and add more details in the supplementary material.
- We restate the reason why **suppressing only the largest singular value** is **not enough** to ensure quantization robustness.
- We argue the superiority of closed-form solution over SGD in terms of **convergence speed, numerical stability, and performance**.
- We restate the algorithm process in detail.
- We provide **extensive experiments** to validate our claims, including comparison with BRECQ, more experiments on EDSR, and skipping on full-rank layers.
- We provide a discussion about the difference between this condition number-based method and outlier-suppression methods.

Thank you once again to all reviewers and area chairs for your time and thoughtful feedback. We hope our responses have addressed your concerns, and would be grateful if these clarifications could assist in forming your final evaluation. Please feel free to let us know if any further clarification is needed.

Best Regards,

Authors

---

### Author Response · Authors · 2025-11-30
**Summary of Previous Rebuttal and Discussion**

Dear Area Chair,

We thank all reviewers (R1-N5vY, R2-X7t9, R3-xoSz, R4-Zbn9) and area chairs for their insightful comments and valuable time. To reduce the workload of the area chairs, we now summarize the reviews, rebuttals and discussions as follows:

## The advantages of our paper

1. Performance: Our paper achieves outstanding performance on five benchmarks, different scale ratios and bit-widths. The SOTA performance is acknowledged by all reviewers (R1, R2, R3, R4). Moreover, the proposed method is practical and lightweight, which is suitable for real-world applications, acknowledged by reviewers R2 and R3
2. Theoretical analysis: The motivation is based on a theoretical discovery while the whole method is based on strong theoretical analysis. The reviewers (R1, R2, R3) are convinced by the theoretical analysis.
3. Organization: The paper is well-organized. The reviewers (R2, R3) are convinced by the organization.

## The weaknesses of our paper

The major concerns of Reviewer N5vY are four points. The first two points are about the choice for solving the optimization problem. We have provided both theoretical analysis and experimental results to justify our choice. The third point is about the pipeline of our method. We have clarified the process of our method in detail. The fourth point is about the task to validate our method. We have provided three convincing reasons for choosing SR for CondiQuant.

Both Reviewer X7t9 and zBn9 have little concern about the whole algorithm, but only about the details of the implementation. We have provided detailed explanations, and the reviewers have accepted our suggestions.

Reviewer xoSz first concern is about motivation of the proposed method. We have provided detailed explanations and analysis to solve the problem. Secondly, Reviewer xoSz concerns the second-order term in the optimization problem. We have provided theoretical analysis to justify that the second-order term is negligible. Thirdly, Reviewer xoSz concerns the generality across architectures. We provide the results on EDSR to demonstrate the excellent performance of the proposed CondiQuant.

## Summary

Our paper has convincing advantages and theoretical analysis, which is widely acknowledged by reviewers. There are some minor concerns about the implementation details, which are well explained with theoretical analysis, experimental results or both. We believe we have addressed all the concerns of reviewers. If there are any further questions, please feel free to contact us.

Best regards,

Authors

---

### Meta-Review · Area_Chair_QNpD · 2026-01-02

**Summary:**

This paper proposes an effective condition number based post-training quantization method for image super-resolution. Reviewers raise lots of concerns of this paper including the generalization of the proposed method, the motivation of the proposed method, and insufficient ablation studies of the proposed component.

**Reviewer Concerns:**

The major concerns of reviewers including the generalization of the proposed method and the motivation of the proposed method should be addressed.

**Reviewer Scores:**

The authors provide rebuttals to solve the concerns of reviewers. For the generalization of the proposed method, the evaluations on the  arbitray-scale image super-resolution's backbone should be provided. In addition, the evaluations on larger SwinIR variants should be provided. However, in the rebuttals, the authors only provide results using the EDSR method.

---

### Decision · Program_Chairs · 2026-01-26

Reject